# Buckling Behavior of FG-CNT Reinforced Composite Conical Shells Subjected to a Combined Loading

**DOI:** 10.3390/nano10030419

**Published:** 2020-02-28

**Authors:** Abdullah H. Sofiyev, Francesco Tornabene, Rossana Dimitri, Nuri Kuruoglu

**Affiliations:** 1Department of Civil Engineering of Engineering Faculty, Suleyman Demirel University, 32260 Isparta, Turkey; 2Department of Innovation Engineering, University of Salento, 73100 Lecce, Italy; francesco.tornabene@unibo.it (F.T.); rossana.dimitri@unisalento.it (R.D.); 3Department of Civil Engineering of Faculty of Engineering and Architecture, Istanbul Gelisim University, 34310 Istanbul, Turkey; nkuruoglu@gelisim.edu.tr

**Keywords:** nanocomposites, buckling, FG-CNTRC, truncated cone, critical combined loads

## Abstract

The buckling behavior of functionally graded carbon nanotube reinforced composite conical shells (FG-CNTRC-CSs) is here investigated by means of the first order shear deformation theory (FSDT), under a combined axial/lateral or axial/hydrostatic loading condition. Two types of CNTRC-CSs are considered herein, namely, a uniform distribution or a functionally graded (FG) distribution of reinforcement, with a linear variation of the mechanical properties throughout the thickness. The basic equations of the problem are here derived and solved in a closed form, using the Galerkin procedure, to determine the critical combined loading for the selected structure. First, we check for the reliability of the proposed formulation and the accuracy of results with respect to the available literature. It follows a systematic investigation aimed at checking the sensitivity of the structural response to the geometry, the proportional loading parameter, the type of distribution, and volume fraction of CNTs.

## 1. Introduction

Conical shells are well known to play a key role in many applications, including aviation, rocket and space technology, shipbuilding and automotive, energy and chemical engineering, as well as industrial constructions. In such contexts, carbon nanotubes (CNTs) have increasingly attracted the attention of engineers and designers for optimization purposes, due to their important physical, chemical, and mechanical properties. The shell structures reinforced with CNTs, indeed, are lightweight and resistant to corrosion and feature a high specific strength, with an overall simplification in their manufacturing, transportation, and installation processes.

In many engineering and building structures, shells are subjected to a simultaneous action of different loads, such as a combined compressive force and external pressure, which can affect significantly their global stability, as observed in the pioneering works [1,2,3,4,5], within a parametric study of the buckling response for homogeneous composite cylindrical and conical shells subjected to a combined loading (CL). Among the novel class of composite functionally graded materials (FGMs), the first studies on the buckling response of FGM-based shells subjected to a CL can be found in [6] and [7], for cylindrical and conical shells, as well as in [8,9,10,11,12,13,14] for different shell geometries and boundary conditions, while considering different theoretical approaches. Moreover, the increased development of nanotechnology has induced a large adoption of nano-scale materials, e.g., CNTs, in many engineering systems and devices, discovered experimentally by Iijima [15] in 1991 during the production of fullerene by arc discharge evaporation. It is known from the literature, indeed, that the generation of CNTs is strictly related to the creation and evaporation of fullerene, which is decomposed into graphene to yield different types of CNTs. The tubes obtained by graphite with the arc-evaporation process become hollow pipes when the graphite layer, i.e., graphene, turns into a cylindrical shape [16,17]. Improving the properties of materials through a reinforcement phase is one of the most relevant topics in modern materials science (metamaterials, heterogeneous materials, architectured materials etc.) [18,19,20]. The outstanding mechanical, electrical, and thermal properties of FG CNTs make them very attractive for many current and future engineering applications, more than conventional carbon fiber reinforced composites [21,22,23]. The modern technology has also allowed a combined use of FGMs and CNTs in various structural elements, which is reflected in the introduction of a great number of advanced theoretical and numerical methods to solve even more complicated problems, with a special focus on mesh-free methods [24,25,26,27,28,29,30,31,32].

Among the available literature, the formulation and solution of the buckling and postbuckling problems of carbon nanotube reinforced composite (CNTRC)-cylindrical shells under a CL, was introduced for the first time by Shen and Xiang [33], followed by Sahmani et al. [34] for composite nanoshells, including the effect of surface stresses at large displacements, and by the instability study in [35] for rotating FG-CNTRC-cylindrical shells. In the literature, however, many works focusing on the buckling behavior of FG-CNTRC-shells consider the separate action of axial or lateral loads, see [36,37,38,39,40,41,42,43,44], whereas limited attention has been paid, up to date, to a CL condition. This aspect is considered in the present work for FG-CNTRC-conical shells, whose problem is solved in a closed form through the Galerkin method. A systematic study is performed to evaluate the sensitivity of the buckling response to the geometry, loading condition, distribution, and volume fraction of the reinforcing CNTs, which could be of great interest for design purposes. 

The paper is structured as follows: in Section 2 we present the basic formulation of the problem, whose governing equations are presented in Section 3, and solved in closed form in Section 4. The numerical results from the parametric investigation are analyzed in Section 5, while the concluding remarks are discussed in Section 6.

## 2. Formulation of the Problem

Let us consider an FG-CNTRC truncated conical shell, with length *L*, half-vertex angle γ, end radii $R_{1}$ and $R_{2}$ (with $R_{1}<R_{2}$), and thickness h, as schematically depicted in Figure 1, along with the displacement components *u*, *v*, and *w* of an arbitrary point at the reference surface. The FG-CNTRC-conical shells (CSs) are subjected to a combined axial compression and uniform external pressures, as follows:(1)$$\begin{array}{ccc} N_{x0}=-T_{ax}-0.5xP_{1}\tan\gamma , & N_{\theta 0}=-xP_{2}\tan\gamma , & N_{x\theta 0}=0 \end{array}$$
where $N_{x0}$, $N_{\theta 0}$, $N_{x\theta 0}$ are the membrane forces for null initial moments, $T_{ax}$ is the axial compression, and $P_{j}(j=1,2)$ stands for the uniform external pressures.

If the external pressures in Figure 1 consider only the lateral pressure, it is $T_{ax}=P_{1}=0$ and $P_{2}=P_{L}$, whereas for a hydrostatic pressure, it is assumed $T_{ax}=0$ and $P_{1}=P_{2}=P_{H}$.

In the following formulation, we consider the volume fraction of CNTs and matrix, denoted by $V_{CN}$ and $V_{m}$, respectively, with normal and shear elastic properties $E_{11}^{CN}$, $E_{22}^{CN}$, $G_{12}^{CN}$, for CNTs, and $E^{m}$, $G^{m}$, for the matrix, and efficiency parameters $\eta_{j}(j=1,2,3)$ for CNTs. Thus, the mechanical properties of CNTRC-CSs can be expressed, according to an improved mixture rule [33], as follows:(2)$$\begin{array}{ccccc} E_{11}=\eta_{1}V_{CN}E_{11}^{CN}+V_{m}E^{m}, & \frac{\eta_{2}}{E_{22}}=\frac{V_{CN}}{E_{22}^{CN}}+\frac{V_{m}}{E^{m}}, & \frac{\eta_{3}}{G_{12}}=\frac{V_{CN}}{G_{12}^{CN}}+\frac{V_{m}}{G^{m}}, & G_{13}=G_{12}, & G_{23}=1.2G_{12} \end{array}$$
where the volume fraction of CNTs and matrix are related as $V_{CN}+V_{m}=1$. 

The volume fraction of the FG-CNTRC-CS is assumed as follows:(3)$$\begin{array}{ll} V_{CN}=\left(1-2\bar{z}\right)V_{CN}^{*}\;\;\mathrm{for}\;\;\mathrm{FG}-V & \\ V_{CN}=\left(1+2\bar{z}\right)V_{CN}^{*}\;\;\mathrm{for}\;\;\mathrm{FG}-\Lambda & \\ V_{CN}=4\left|\bar{z}\right|V_{CN}^{*}\;\;\mathrm{for}\;\;\mathrm{FG}-X, & \bar{z}=z/h \end{array}$$
where $V_{CN}^{*}$ is the volume fraction of the CNT, expressed as
(4)$$V_{CN}^{*}=\frac{w_{CN}}{w_{CN}+(\rho^{CN}/\rho^{m})-(\rho^{CN}/\rho^{m})w_{CN}}$$
whereby the mass fraction of CNTs is denoted by $w_{CN}$, and the density of CNTs and matrix are defined as $\rho^{CN}$ and $\rho^{m}$, respectively. In our case, for a uniform distribution (UD)-CNTRC-CSs, it is $V_{CN}=V_{CN}^{*}$. The Poisson’s ratio is defined as
(5)$$\mu_{12}=V_{CN}^{*}\mu_{12}^{CN}+V_{m}\mu^{m}$$

The topologies of UD- and FG-CNTRC-CSs are shown in Figure 2.

## 3. The Governing Equations

Based on the first order shear deformation theory (FSDT), the constitutive stress–strain relations for FG-CNTRC-CSs are expressed as follows: (6)$$\left[\begin{array}{c} \tau_{x}\\ \tau_{\theta }\\ \tau_{xz}\\ \tau_{\theta z}\\ \tau_{x\theta } \end{array}\right]=\left[\begin{array}{ccccc} {\bar{E}}_{11}\left(\bar{z}\right) & {\bar{E}}_{12}\left(\bar{z}\right) & 0 & 0 & 0\\ {\bar{E}}_{21}\left(\bar{z}\right) & {\bar{E}}_{22}\left(\bar{z}\right) & 0 & 0 & 0\\ 0 & 0 & {\bar{E}}_{44}\left(\bar{z}\right) & 0 & 0\\ 0 & 0 & 0 & {\bar{E}}_{55}\left(\bar{z}\right) & 0\\ 0 & 0 & 0 & 0 & {\bar{E}}_{66}\left(\bar{z}\right) \end{array}\right]\left[\begin{array}{c} \varepsilon_{x}\\ \varepsilon_{\theta }\\ \gamma_{xz}\\ \gamma_{\theta z}\\ \gamma_{x\theta } \end{array}\right]$$
where $\tau_{ij}(i,j=x,\theta ,z)$, $\varepsilon_{jj}(j=x,\theta )$, and $\gamma_{ij}(i,j=x,\theta ,z)$ are the stress and strain tensors of FG-CNTRC-CSs, respectively, and the coefficients $E_{ij}\left(\bar{z}\right)$, (i,j=1,2,6) are defined as
(7)$$\begin{array}{l} {\bar{E}}_{11}(\bar{z})=\frac{E_{11}(\bar{z})}{1-\mu_{12}\mu_{21}},{\bar{E}}_{22}(\bar{z})=\frac{E_{22}(\bar{z})}{1-\mu_{12}\mu_{21}(\bar{z})}\\ {\bar{E}}_{12}(\bar{z})=\frac{\mu_{21}E_{11}(\bar{z})}{1-\mu_{12}\mu_{21}}=\frac{\mu_{12}E_{22}(\bar{z})}{1-\mu_{12}\mu_{21}}={\bar{E}}_{21}(\bar{z}),\\ {\bar{E}}_{44}(\bar{z})=G_{23}(\bar{z}),{\bar{E}}_{55}(\bar{z})=G_{13}(\bar{z}),{\bar{E}}_{66}(\bar{z})=G_{12}(\bar{z}) \end{array}$$

The shear stresses of FG-CNTRC-CSs vary throughout the thickness direction as follows [45,46]:(8)$$\begin{array}{ccc} \tau_{z}=0, & \tau_{xz}=\frac{du_{1}(z)}{dz}\varphi_{1}(x,\theta ), & \tau_{\theta z}=\frac{du_{2}(z)}{dz}\varphi_{2}(x,\theta ) \end{array}$$
where $\varphi_{1}(x,\theta )$ and $\varphi_{2}(x,\theta )$ are the rotations of the reference surface about the θ and x axes, respectively, and $u_{1}\left(z\right)$ and $u_{2}\left(z\right)$ refer to the shear stress shape functions.

By combining Equations (6) and (8), we get the following strain relationships:(9)$$\left[\begin{array}{l} \varepsilon_{x}\\ \varepsilon_{\theta }\\ \gamma_{x\theta } \end{array}\right]=\left[\begin{array}{l} e_{x}-z\frac{\partial^{2}w}{\partial x^{2}}+F_{1}(z)\frac{\partial \varphi_{1}}{\partial x}\\ e_{\theta }-z\left(\frac{1}{x^{2}}\frac{\partial^{2}w}{\partial \alpha^{2}}+\frac{1}{x}\frac{\partial w}{\partial x}\right)+F_{2}(z)\frac{1}{x}\frac{\partial \varphi_{2}}{\partial \alpha }\\ \gamma_{0x\theta }-2z\left(\frac{1}{x}\frac{\partial^{2}w}{\partial x\partial \alpha }-\frac{1}{x^{2}}\frac{\partial w}{\partial \alpha }\right)+F_{1}(z)\frac{1}{x}\frac{\partial \varphi_{1}}{\partial \alpha }+F_{2}(z)\frac{\partial \varphi_{2}}{\partial x} \end{array}\right]$$
where α=θsinγ and $e_{x}$, $e_{\theta }$, $\gamma_{0x\theta }$ stand for the strain components at the reference surface, and $F_{1}\left(z\right)$, $F_{2}\left(z\right)$ are defined as
(10)$$\begin{array}{cc} F_{1}(z)=\int_{0}^{z}\frac{1}{{\bar{E}}_{55}(\bar{z})}\frac{du_{1}}{dz}dz, & F_{2}(z)=\int_{0}^{z}\frac{1}{{\bar{E}}_{44}(\bar{z})}\frac{du_{1}}{dz}dz \end{array}$$

The internal actions can be defined in approximate form as follows [46,47,48]:(11)$$\left(N_{ij},Q_{i},M_{ij}\right)=\int_{-h/2}^{h/2}\left(\tau_{ij},\tau_{iz},z\tau_{ij}\right)dz,\;\;(i,j=x,\theta )$$

By introducing the Airy stress function (Φ) satisfying [[45],[47],], Equation (11) becomes as follows:(12)$$\left(N_{x},N_{\theta },N_{x\theta }\right)=h\left[\frac{1}{x}\left(\frac{1}{x}\frac{\partial^{2}}{\partial \alpha^{2}}+\frac{\partial }{\partial x}\right),\frac{\partial^{2}}{\partial x^{2}},-\frac{1}{x}\left(\frac{\partial^{2}}{\partial x\partial \alpha }-\frac{1}{x}\frac{\partial }{\partial \alpha }\right)\right]\Phi$$

By using Equations (6), (9), (11), and (12), we obtain the expressions for force, moment, and strain components in the reference surface, which are then substituted in the stability and compatibility equations [45,47] to obtain the following governing differential equations for FG-CNTRC-CSs under a CL, with independent parameters Φ, w, $\varphi_{1}$, $\varphi_{2}$, i.e.,
(13)$$\begin{array}{l} L_{11}\Phi +L_{12}w+L_{13}\varphi_{1}+L_{14}\varphi_{2}=0\\ L_{21}\Phi +L_{22}w+L_{23}\varphi_{1}+L_{24}\varphi_{2}=0\\ L_{31}\Phi +L_{32}w+L_{33}\varphi_{1}+L_{34}\varphi_{2}=0\\ L_{41}\Phi +L_{42}w+L_{43}\varphi_{1}+L_{44}\varphi_{2}=0 \end{array}$$
where $L_{ij}(i,j=1,2,3,4)$ are differential operators, whose details are described in Appendix A.

## 4. Solution Procedure 

The approximating functions for conical shells with free supports are assumed as
(14)$$\begin{array}{l} \begin{array}{cc} \Phi =\bar{\Phi }x_{2}e^{(a+1)\bar{x}}\sin(n_{1}\bar{x})\cos(n_{2}\alpha ), & w=\bar{w}e^{a\bar{x}}\sin(n_{1}\bar{x})\cos(n_{2}\alpha ), \end{array}\\ \begin{array}{cc} \varphi_{1}={\bar{\varphi }}_{1}e^{a\bar{x}}\cos(n_{1}\bar{x})\cos(n_{2}\alpha ), & \varphi_{2}={\bar{\varphi }}_{2}e^{a\bar{x}}\sin(n_{1}\bar{x})\sin(n_{2}\alpha ) \end{array} \end{array}$$
where $\bar{\Phi }$, $\bar{w}$, ${\bar{\varphi }}_{1}$, ${\bar{\varphi }}_{2}$ are the unknown constants, a is an unknown coefficient to be determined with the enforcement of the minimum conditions for combined buckling loads, $\bar{x}=\ln\left(\frac{x}{x_{2}}\right)$, $n_{1}=\frac{m\pi }{x_{0}}$, $n_{2}=\frac{n}{\sin\gamma }$, $x_{0}=\ln\left(\frac{x_{2}}{x_{1}}\right)$, with *m* and *n* the wave numbers.

By introducing Equation (14) into Equation (13), and by some manipulation and integration, we determine the nontrivial solution by enforcing
(15)$$\det(c_{ij})=0$$
where $c_{ij}(i,j=1,2,\ldots ,4)$ are the matrix coefficients, as defined in Appendix B. 

Equation (15) can be rewritten in expanded form as follows:(16)$$c_{41}\Gamma_{1}-\left(T_{ax}c_{T}+P_{1}c_{P_{1}}+P_{2}c_{P_{2}}\right)\Gamma_{2}+c_{43}\Gamma_{3}+c_{44}\Gamma_{4}=0$$
where $c_{T}$ is the axial load parameter, $c_{P_{1}}$ and $c_{P_{2}}$ are the external pressure parameters, whose expression are given in Appendix B, while parameters $\Gamma_{j}(j=1,2,3,4)$ are defined as
(17)$$\begin{array}{cccc} \Gamma_{1}=-\left|\begin{array}{lll} c_{12} & c_{13} & c_{14}\\ c_{22} & c_{23} & c_{24}\\ c_{32} & c_{33} & c_{34} \end{array}\right|, & \Gamma_{2}=\left|\begin{array}{lll} c_{11} & c_{13} & c_{14}\\ c_{21} & c_{23} & c_{24}\\ c_{31} & c_{33} & c_{34} \end{array}\right|, & \Gamma_{3}=-\left|\begin{array}{lll} c_{11} & c_{12} & c_{14}\\ c_{21} & c_{22} & c_{24}\\ c_{31} & c_{32} & c_{34} \end{array}\right|, & \Gamma_{4}=\left|\begin{array}{lll} c_{11} & c_{12} & c_{13}\\ c_{21} & c_{22} & c_{23}\\ c_{31} & c_{32} & c_{33} \end{array}\right| \end{array}$$

For FG-CNTRC-CSs under an axial load, it is $P_{1}=P_{2}=0$, while $T_{1SDT}^{axcr}$ is defined as follows:(18)$$T_{1SDT}^{axcr}=\frac{c_{41}\Gamma_{1}+c_{43}\Gamma_{3}+c_{44}\Gamma_{4}}{\Gamma_{2}E_{c}hc_{T}}$$

Differently, for FG-CNTRC-CSs under a uniform lateral pressure, it is $T_{ax}=P_{1}=0$; $P_{2}=P_{L}$, whereas the expression for $P_{1SDT}^{Lcr}$ based on the FSDT is as follows:(19)$$P_{1SDT}^{Lcr}=\frac{c_{41}\Gamma_{1}+c_{43}\Gamma_{3}+c_{44}\Gamma_{4}}{\Gamma_{2}E_{c}c_{P_{L}}}$$

For FG-CNTRC-CSs under a uniform hydrostatic pressure, it is $T_{ax}=0$, $P_{1}=P_{2}=P_{H}$, and $P_{1SDT}^{Hcr}$ is defined as
(20)$$P_{1SDT}^{Hcr}=\frac{c_{41}\Gamma_{1}+c_{43}\Gamma_{3}+c_{44}\Gamma_{4}}{\Gamma_{2}E_{c}c_{PH}}$$
where $c_{PH}$ is a parameter depending on the hydrostatic pressure, as defined in Appendix B. 

For a combined axial load/lateral pressure, and a combined axial load/hydrostatic pressure acting on an FG-CNTRC-CS based on the FSDT, the following relation can be used [47]:(21)$$\frac{T_{1}}{T_{1SDT}^{axcr}}+\frac{P_{1L}}{P_{1SDT}^{Lcr}}=1$$
and
(22)$$\frac{T_{1}}{T_{1SDT}^{axcr}}+\frac{P_{1H}}{P_{1SDT}^{Hcr}}=1$$
where
(23)$$T_{1}=T/E_{c}h,P_{1L}=P_{L}/E_{c},P_{1H}=P_{H}/E_{c}$$

Under the assumptions $T_{1}=\eta P_{1L}$ and $T_{1}=\eta P_{1H}$, in Equations (21) and (22), we get the following expressions: (24)$$P_{1SDT}^{Lcbcr}=\frac{T_{1SDT}^{axcr}P_{1SDT}^{Lcr}}{\eta P_{1SDT}^{Lcr}+T_{1SDT}^{axcr}}$$
and
(25)$$P_{1SDT}^{Hcbcr}=\frac{T_{1SDT}^{axcr}P_{1SDT}^{Hcr}}{\eta P_{1SDT}^{Hcr}+T_{1SDT}^{axcr}}$$
where η≥0 is the dimensionless load-proportional parameter.

From Equations (24) and (25), we obtained the expressions for the critical loads $T_{1CST}^{axcr}$, $P_{1CST}^{Lcr}$, $P_{1CST}^{Hcr}$, $P_{1CST}^{Lcbcr}$, $P_{1CST}^{Hcbcr}$, while neglecting the shear strains.

## 5. Results and Discussion

### 5.1. Introduction

In this section, a poly methyl methacrylate (PMMA) reinforced with (10,10) armchair Single Walled CNTs (SWCNTs) was considered for the numerical investigation. The effective material properties of CNTs and PMMA matrix are reported in Table 1 (see [46]), along with the efficiency parameters for three volume fractions of CNTs. The shear stress quadratic shape functions are distributed as $u_{1}\left(z\right)=u_{2}\left(z\right)=z-4z^{3}/3h^{2}$ [46]. The critical CL values for FG-CNTRC-CSs are determined for different magnitudes of *a*, within a coupled stress theory (CST) context, in order to check for the effect of the FSDT on the critical loading condition. After a systematic numerical computation, it is found that for freely supported FG-CNTRC-CSs, the critical values of a CL were reached for a=2.4.

### 5.2. Comparative Evaluation

As a first comparative check, the critical lateral pressure, axial load, and combined load of shear deformable CNTRC-CYLSs with an FG-X profile is evaluated as in [33]. The CNTRC-CS reverts to a CNTRC-CYLS, when γ tends to zero. The CNTRC-CYLS has radius r and length $L_{1}$ with the following geometrical properties: r/h=30, h=2 mm, and $L_{1}=\sqrt{300rh}$, whereas the material properties are assumed as in Table 1, for *T* = 300 K, see [33]. The magnitudes of the critical loading for CNTRC-CYLSs were obtained for *a* = 0. Based on Table 2, a good agreement between our results and predictions by Shen and Xiang [33] is observable for the critical lateral pressure, axial load, and CL.

In Table 3, the values of $P_{SDT}^{Lcr}$ of FG-CNTRC-CSs with different profiles and half-vertex angles are compared with results by [37], based on the GDQ method. A FG-CNTRC is considered for $V_{CN}^{*}=0.17$, $L=\sqrt{300R_{1}h}$, $R_{1}/h=100$, and h=1 mm. Based on Table 3, the correspondence between our values of $P_{SDT}^{Lcr}$ and predictions by Jam and Kiani [37], verifies the consistency of our formulation. 

### 5.3. Analysis of Combined Buckling Loads

In what follows, we analyze the sensitivity of the critical loading to FG profiles, volume fractions of CNT, and FSDT formulation, by considering the ratios $100\%\times \frac{P_{FG-CN}^{cbcr}-P_{UD}^{cbcr}}{P_{UD}^{cbcr}}$ and $100\%\times \frac{P_{1CST}^{cbcr}-P_{1SDT}^{cbcr}}{P_{1CST}^{cbcr}}$. One of the main parameters affecting a critical CL is represented by the load-proportional parameter. In Figure 3 and Figure 4, we plot the variations of $P_{1SDT}^{Lcbcr}$, $P_{1CST}^{Lcbcr}$, as shown in Figure 3, $P_{1SDT}^{Hcbcr}$ and $P_{1CST}^{Hcbcr}$, as shown in Figure 4, vs. the dimensionless load-proportional parameter η, for UD- and FG-CNTRCs. Based on both figures, for all profiles, the magnitudes of the critical CL decrease for an increasing dimensionless load-proportional parameter η. This sensitivity is more pronounced for a FG−Λ profile, compared to a FG-V or FG-X profile. The strong influence of the FG profiles, *V_CN_*, and shear strains on the critical CLs depends on the dimensionless load-proportional parameter. 

A pronounced sensitivity of $P_{1SDT}^{Lcbcr}$ and $P_{1SDT}^{Hcbcr}$ to the FG-CNTRC types, was noticed for η ranging between 100 and 800, whereas a certain influence was observed in a horizontal sense, as η>800, for a fixed $V_{CN}^{*}$. This last phenomenon can be explained by the fact that, for large values of η, the axial load prevails over the external pressure. For a fixed value of $V_{CN}^{*}=0.12$, the effect of a FG profile (i.e., FG-V, FG−Λ, FG-X, respectively) on the $P_{1SDT}^{Lcbcr}$ is estimated as (−10%), (−14.9%), (+14.4%) for *η* = 100; (−17%), (−21.4%), (+21.8%) for *η* = 800; (−17.04%), (−21.4%), (+21.9%) for *η* = 1000. Compared to a uniform distribution (UD), the highest sensitivity of $P_{1SDT}^{Lcbcr}$ is noticed for $V_{CN}^{*}=0.28$ in the FG-X profile, whereby a FG-V type distribution features the lowest sensitivity for the same fixed value of $V_{CN}^{*}=0.28$. 

A small influence of the shear strain is observable for an increasing value of η and a FG-X profile. This influence continues to decrease by almost 2%–3%, for a FG-V and FG−Λ profile with different $V_{CN}^{*}$, while reaching the lowest percentage at $V_{CN}^{*}$ = 0.17 for a FG−Λ profile.

Table 4 summarizes the variation of the critical CLs, based on a FSDT and CST, for a FG-V and FG-X profile, and different $R_{1}/h$ ratios. The geometrical data for numerical computations are provided in the same table. Please note that the critical CL decreases monotonically, along with a gradual increase of the circumferential wave numbers, for an increased value of $R_{1}/h$.

An irregular effect of the FG-V and FG-X profile on the $P_{1SDT}^{Lcbcr}$ and $P_{1SDT}^{Hcbcr}$ is observed with $R_{1}/h$, for a fixed $V_{CN}^{*}$. When the dimensionless $R_{1}/h$ ratio increases from 30 up to 90, the influence of an FG-V profile on the CL values tends to decrease, whereas the effect of a FG-X profile on the CL increases, for an increasing value of $R_{1}/h$ from 30 to 70. After this value, the effect decreases slightly. The shear strains reduce significantly the influence of FG-V and FG-X profiles on both CLs. For example, the effect of a FG-V profile on $P_{1SDT}^{Lcbcr}$ is less pronounced than $P_{1CST}^{Lcbcr}$ by about 3%–12%. This difference becomes meaningful and varies from 3% up to 23% for a FG-X profile, depending on the selected $R_{1}/h$ ratio. Note that the highest FG effect on the $P_{1SDT}^{Lcbcr}$ occurs for a FG-X profile (+35.47%), at $R_{1}/h$= 90 and $V_{CN}^{*}$ = 0.28, while the lowest effect (−11.69%) is found for a FG-V profile, at $R_{1}/h$ = 90 and $V_{CN}^{*}$ = 0.17. These effects are more pronounced for a combined axial load and hydrostatic pressure ($P_{1SDT}^{Hcbcr}$), compared to a combined axial load and lateral pressure ($P_{1SDT}^{Lcbcr}$). The influence of the shear strain on $P_{1SDT}^{Lcbcr}$ for an FG-X and FG-V profile decreases for each fixed value of $V_{CN}^{*}$, and remains significant for an increased value of $R_{1}/h$ up to 90. A similar sensitivity to the shear strain was observed for $P_{1SDT}^{Hcbcr}$, but was less pronounced than the one for $P_{1SDT}^{Lcbcr}$. The highest shear strain effect on $P_{1SDT}^{Lcbcr}$ is observed for an FG-X profile (-52.8%), at $R_{1}/h$ = 30 and $V_{CN}^{*}$ = 0.28, whereas the lowest shear strain influence on $P_{1SDT}^{Hcbcr}$ is noticed for an FG-V profile (+1.54%), at $R_{1}/h$ = 70 and $V_{CN}^{*}$ = 0.28. An increased value of $V_{CN}^{*}$ yields an irregular effect of the shear strains on the critical CL. 

The variation of the critical CL for UD- and FG-CNTRC-CSs with different profiles is plotted in Figure 5 and Figure 6, vs. γ. It seems that the critical value of the CL decreases for an increased value of γ. As γ increases from 15° to 60°, the effect of FG-V and FG-X distributions on the critical magnitude of the CL decreases slightly, whereby it decreases rapidly as $\gamma >60^{\circ }$ for all values of $V_{CN}^{*}$. Furthermore, the effect of CNT distribution on $P_{1SDT}^{Lcbcr}$ and $P_{1SDT}^{Hcbcr}$ maintains almost the same for different γ. The shear strain effect on the critical CL depends on the selected CNT profile, especially for a FG-X profile. A remarkable shear strain influence of (+55.56%) on the critical value of the CL occurs at $V_{CN}^{*}=$ 0.28 and γ = 75°.

## 6. Conclusions

The buckling of FG-CNTRC-CSs subjected to a combined loading was here studied based on a combined Donnell-type shell theory and FSDT. The FG-CNTRC-CS properties were assumed to vary gradually in the thickness direction with a linear distribution of the volume fraction *V_CN_* of CNTs. The governing equations were converted into algebraic equations using the Galerkin procedure, and the analytical expression for the critical value of the combined loading was found. The solutions were compared successfully with results in the open literature, thus confirming the accuracy of the proposed formulation. A novel buckling analysis was, thus, performed for both a uniform distribution and FG distribution of CNTs, while determining the effect of the volume fraction and shell geometry on the critical value of the combined loading condition, as useful for practical engineering applications.

## Figures and Tables

**Figure 1 nanomaterials-10-00419-f001:**
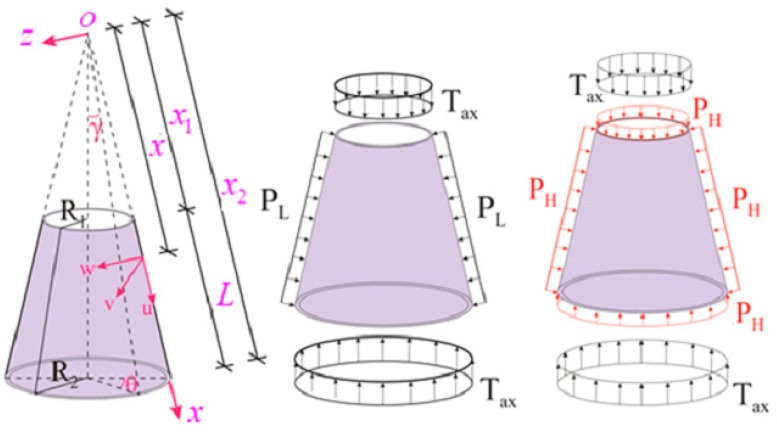
The functionally graded carbon nanotube reinforced composite conical shell (FG-CNTRC-CS) subjected to a combined loading (CL).

**Figure 2 nanomaterials-10-00419-f002:**
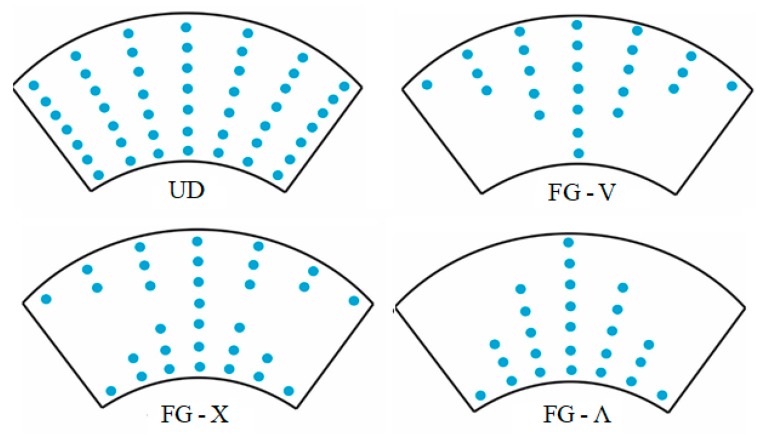
Configurations of uniform distribution (UD)-CNTRC-CSs and three types of FG-CNTRC-CSs.

**Figure 3 nanomaterials-10-00419-f003:**
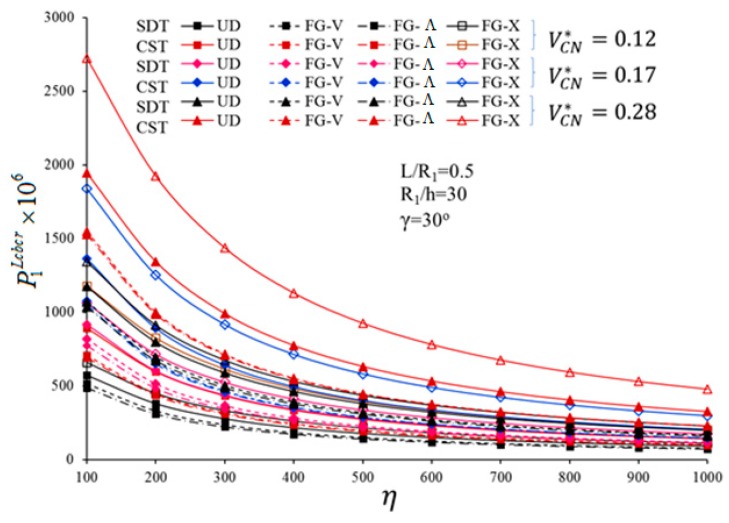
Variation of $P_{1SDT}^{Lcbcr}$ and $P_{1CST}^{Lcbcr}$ for UD- and FG-CNTRC-CSs with different $V_{CN}^{*}$ and load-proportional parameter η.

**Figure 4 nanomaterials-10-00419-f004:**
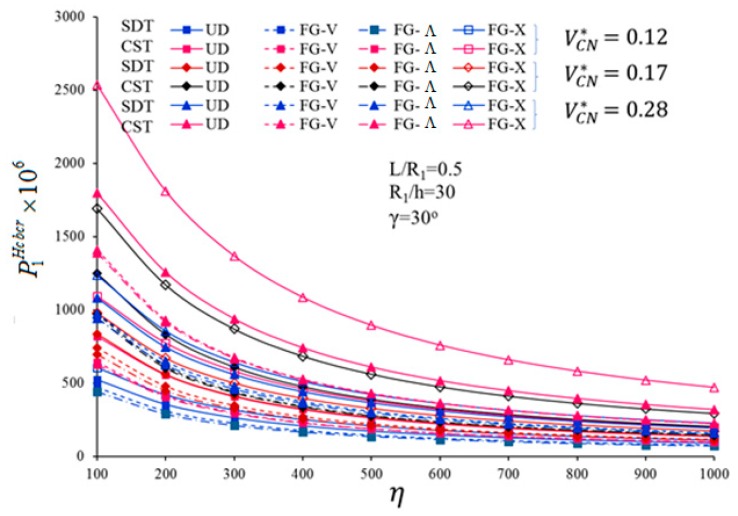
Variation of $P_{1SDT}^{Hcbcr}$ and $P_{1CST}^{Hcbcr}$ for UD- and FG-CNTRC-CSs with a different $V_{CN}^{*}$ and load-proportional parameter η.

**Figure 5 nanomaterials-10-00419-f005:**
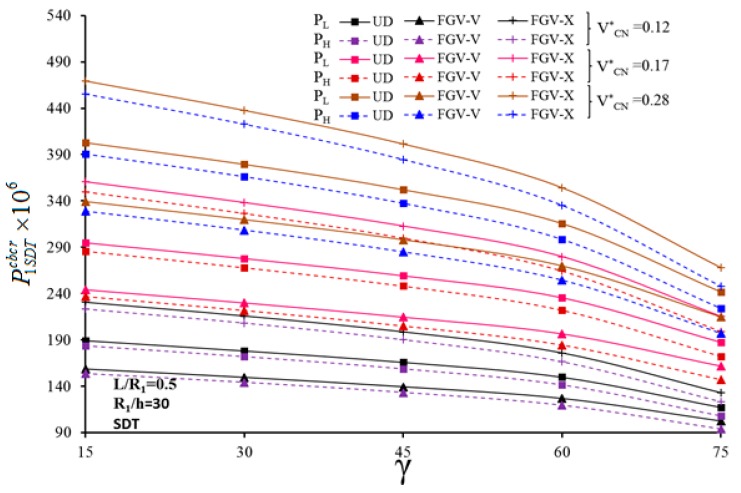
Variation of CCLs for UD- and FG-CNTRC-CSs based on the FSDT with a different $V_{CN}^{*}$ and half-vertex angle γ, η=500.

**Figure 6 nanomaterials-10-00419-f006:**
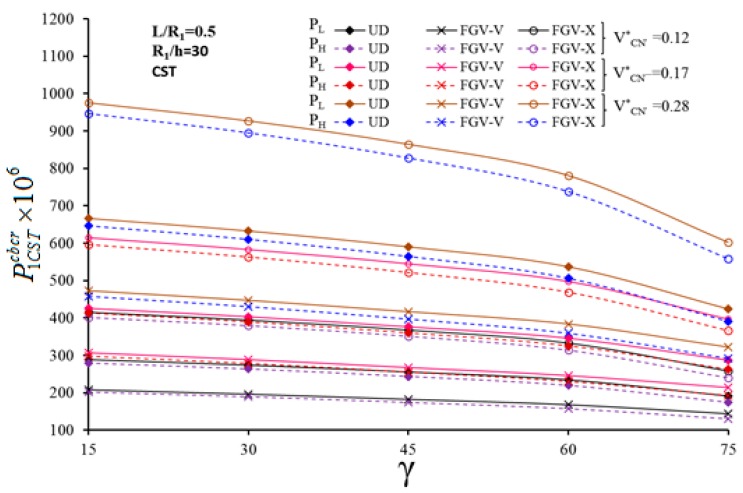
Variation of CCLs for UD- and FG-CNTRC-CSs based on the CST with a different $V_{CN}^{*}$ and half-vertex angle γ, η=500.

**Table 1 nanomaterials-10-00419-t001:** Properties of CNTs and matrix.

	CWCNT	Matrix (PMMA)
Geometrical properties	$\tilde{L}=9.26\;\mathrm{nm}$, $\tilde{r}=0.68\;\mathrm{nm}$, $\tilde{h}=0.067\;\mathrm{nm}$	
Material properties	$E_{11}^{CN}=5.6466\;\mathrm{TPa},E_{22}^{CN}=7.0800\;\mathrm{TPa}$ $G_{12}^{CN}=1.9445\;\mathrm{TPa},$ $\mu_{12}^{CN}=0.175;\rho^{CN}=1400\;\mathrm{kg}/m^{3}$	$E^{m}=2.5\;\mathrm{Pa},$ $\mu^{m}=0.34,$ $\rho^{m}=1150\;\mathrm{kg}/m^{3}$
CNT efficiency parameter	$\eta_{1}=0.137,\eta_{2}=1.022,\eta_{3}=0.715$ for $V_{CN}^{*}=0.12$; $\eta_{1}=0.142,\eta_{2}=1.626,\eta_{3}=1.138$ for $V_{CN}^{*}=0.17$; $\eta_{1}=0.141,\eta_{2}=1.585,\eta_{3}=1.109$ for $V_{CN}^{*}=0.28$.	

**Table 2 nanomaterials-10-00419-t002:** Comparative response of shear deformable CNTRC-CYLSs with the FG-X profile under a separate or combined axial load and lateral pressure.

	$T_{SDT}^{axcr}\;(\mathrm{MPa})$	$P_{SDT}^{Lcr}(\mathrm{MPa})$	$P_{SDT}^{Lcbcr}(\mathrm{MPa})$	
			*η* = 750	*η* = 140
$V_{CN}^{*}$	Shen and Xiang [33]			
0.12	118.848	0.285	0.112	0.218
0.17	196.376	0.484	0.190	0.370
0.28	247.781	0.616	0.242	0.470
	Present study			
0.12	117.840	0.281	0.111	0.2181
0.17	197.515	0.479	0.188	0.3711
0.28	247.062	0.613	0.2414	0.4756

**Table 3 nanomaterials-10-00419-t003:** Comparative response of shear deformable CNTRC-CSs with the different profiles under lateral pressure for a different half-vertex angle.

	$P_{SDT}^{Lcr}\;(\mathrm{in}\;\mathrm{kPa}),\;(n_{\mathrm{cr}})$			
	*γ*	10°	20°	30°
Jam and Kiani [37]	UD	31.11(8)	24.31(9)	19.00(9)
	FG-X	34.53(8)	27.24(9)	21.38(9)
	FG-V	32.41(8)	25.19(9)	19.52(10)
Present study	UD	31.01(8)	23.91(9)	18.23(10)
	FG-X	34.38(8)	26.69(9)	20.49(10)
	FG-V	32.40 (8)	24.97 (9)	19.77 (10)

**Table 4 nanomaterials-10-00419-t004:** Variation of the critical CLs for UD- and FG-CNTRC-CSs based on first order shear deformation theory (FSDT) and coupled stress theory (CST) with different $V_{CN}^{*}$ and $R_{1}/h$ ratios. $L/R_{1}=0.5$, $\gamma =30^{{}^{\circ}}$, η=500.

$V_{CN}^{*}$	*R*_1_/*h*	Types	$P_{1SDT}^{cbLcr}\;(n_{\mathrm{cr}})$	$P_{1CST}^{cbLcr}(n_{\mathrm{cr}})$	$P_{1SDT}^{cbHcr}(n_{\mathrm{cr}})$	$P_{1CST}^{cbHcr}(n_{\mathrm{cr}})$
0.12	30	UD	178.227(8)	273.320(7)	172.051(8)	263.619(6)
		FGV-V	149.371(7)	196.380(5)	144.088(7)	189.196(5)
		FGV-X	215.601(9)	393.913(8)	208.297(9)	380.032(7)
	50	UD	83.296(9)	96.267(9)	78.996(9)	91.297(9)
		FGV-V	68.684(8)	72.655(8)	65.013(8)	68.772(8)
		FGV-X	107.856(10)	134.647(10)	102.492(10)	127.951(10)
	70	UD	44.157(11)	46.897(11)	41.493(11)	44.068(11)
		FGV-V	37.515(10)	37.035(10)	35.124(9)	34.636(9)
		FGV-X	57.660(12)	63.714(12)	54.360(12)	60.067(12)
	90	UD	26.050(12)	26.743(12)	24.320(12)	24.967(12)
		FGV-V	22.882(11)	21.902(11)	21.266(11)	20.355(11)
		FGV-X	33.770(13)	35.488(13)	31.663(13)	33.274(13)
0.17	30	UD	277.753(7)	403.421(6)	267.928(7)	388.892(6)
		FGV-D	216.472(7)	285.441(6)	208.815(7)	275.161(6)
		FGV-X	337.997(8)	582.930(7)	326.283(8)	562.312(7)
	50	UD	127.380(9)	144.108(9)	120.804(9)	136.614(8)
		FGV-V	104.824(8)	108.979(8)	99.222(8)	103.074(7)
		FGV-X	166.556(10)	202.672(9)	157.999(9)	192.209(9)
	70	UD	67.718(10)	71.100(10)	63.421(10)	66.589(10)
		FGV-V	57.633(9)	56.330(9)	53.797(9)	52.580(9)
		FGV-X	89.216(11)	97.309(11)	83.834(11)	91.439(11)
	90	UD	40.167(12)	40.992(12)	37.468(11)	38.204(11)
		FGV-V	35.470(11)	33.809(11)	32.965(11)	31.366(10)
		FGV-X	52.637(13)	54.879(13)	49.194(12)	51.292(12)
0.28	30	UD	379.351(8)	632.214(7)	366.204(8)	609.852(7)
		FGV-V	319.603(7)	446.857(6)	308.298(7)	430.763(6)
		FGV-X	437.545(9)	927.004(8)	422.721(9)	894.878(8)
	50	UD	182.000(10)	218.225(10)	172.950(10)	207.373(10)
		FGV-V	148.010(9)	162.024(8)	140.108(8)	153.364(8)
		FGV-X	234.684(10)	314.925(11)	223.013(10)	299.268(10)
	70	UD	96.385(12)	104.268(12)	90.694(11)	98.163(11)
		FGV-V	79.650(10)	80.897(10)	74.597(10)	75.765(10)
		FGV-X	128.856(12)	148.054(12)	121.480(12)	139.579(12)
	90	UD	56.322(13)	58.452(13)	52.808(13)	54.805(13)
		FGV-V	47.930(12)	47.203(11)	44.628(11)	43.870(11)
		FGV-X	76.299(13)	82.031(14)	71.538(13)	76.978(13)

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
