# Peer review of "Buckling Behavior of FG-CNT Reinforced Composite Conical Shells Subjected to a Combined Loading"

_nanomaterials, 2020, doi:10.3390/nano10030419_

Round 1
Reviewer 1 Report
The manuscript seem to overall be sound and suitable and I have no substantial comments. Nevertheless, I have some general comments, which I would recommend the Authors to reflect upon.
I suggest to Authors not to use acronyms in the title of the article, but their extension because this would help the comprehension of future readers in this field.
When Authors declare in line 52 the outstanding mechanical, electrical and thermal properties of FG CNTs, a clarification and/or deeper explanation is dutiful in the discussion on the effect of thermal properties in relation to buckling behaviour (see reference: Mirzaei M., Kiani Y., Thermal buckling of temperature dependent FG-CNT reinforced composite conical shells. Aerospace Science and Technology 2015; 47:42-53.)
Therefore, I would like suggest the Authors to rewrite the introduction stressing what is new and what is the emerging.
Furthermore, the Authors cannot fail to mention in the introduction among lines 52-57 also the following references [1-3] because other Authors proposed solution for the thermal buckling and post-buckling of piezoelectric FG-CNTs. In this specific case, I would like suggest to mention that the mechanical, electrical, and thermal properties of FG CNTs are also attractive more than piezoelectric ZnO nanowires, see reference [4].
- Belal Ahmed Mohamed M. S., Zishun L., Kim Meow L. Active control of functionally graded carbon nanotube-reinforced composite plates with piezoelectric layers subjected to impact loading. Journal of Vibration and Control 2019, vol.0:0; 1-18.
- Fan, J. Huang, J. Ding, J. Zhang. Free vibration of functionally graded carbon nanotube-reinforced conical panels integrated with piezoelectric layers subjected to elastically restrained boundary conditions. Advances in Mechanical Engineering 2017, vol.9:7; 1-17.
- Rafiee M., Yang J. and Kitipornchai S. Thermal bifurcation buckling of piezoelectric carbon nanotube reinforced composite beams. Comput Math Appl 2013; 66: 1147–1160.
- Araneo R., Bini F., Rinaldi A., Notargiacomo A., Pea M., Celozzi S. Thermal-electric model for piezoelectric ZnO nanowires. Nanotechnology 2015 vol.26: 26; 265402.
I trust Authors will make the appropriate decision regarding these suggestions.
Author Response
19.02.2020
EXPLANATION TO REVIEWER 1:
First of all, we would like to thank the highly respected Reviewer 1 for his/her improving remarks and the time spent for them.
Comments and Suggestions for Authors
-Reviewer 1
The manuscript seem to overall be sound and suitable and I have no substantial comments. Nevertheless, I have some general comments, which I would recommend the Authors to reflect upon.
I suggest to Authors not to use acronyms in the title of the article, but their extension because this would help the comprehension of future readers in this field.
When Authors declare in line 52 the outstanding mechanical, electrical and thermal properties of FG CNTs, a clarification and/or deeper explanation is dutiful in the discussion on the effect of thermal properties in relation to buckling behaviour (see reference: Mirzaei M., Kiani Y., Thermal buckling of temperature dependent FG-CNT reinforced composite conical shells. Aerospace Science and Technology 2015; 47:42-53.)
Therefore, I would like suggest the Authors to rewrite the introduction stressing what is new and what is the emerging.
Furthermore, the Authors cannot fail to mention in the introduction among lines 52-57 also the following references [1-3] because other Authors proposed solution for the thermal buckling and post-buckling of piezoelectric FG-CNTs. In this specific case, I would like suggest to mention that the mechanical, electrical, and thermal properties of FG CNTs are also attractive more than piezoelectric ZnO nanowires, see reference [4].
- Belal Ahmed Mohamed M. S., Zishun L., Kim Meow L. Active control of functionally graded carbon nanotube-reinforced composite plates with piezoelectric layers subjected to impact loading. Journal of Vibration and Control 2019, vol.0:0; 1-18.
- Fan, J. Huang, J. Ding, J. Zhang. Free vibration of functionally graded carbon nanotube-reinforced conical panels integrated with piezoelectric layers subjected to elastically restrained boundary conditions. Advances in Mechanical Engineering 2017, vol.9:7; 1-17.
- Rafiee M., Yang J. and Kitipornchai S. Thermal bifurcation buckling of piezoelectric carbon nanotube reinforced composite beams. Comput Math Appl 2013; 66: 1147–1160.
- Araneo R., Bini F., Rinaldi A., Notargiacomo A., Pea M., Celozzi S. Thermal-electric model for piezoelectric ZnO nanowires. Nanotechnology 2015 vol.26: 26; 265402.
I trust Authors will make the appropriate decision regarding these suggestions.
EXPLANATION: Thank you for your constructive comments and suggestions. All suggested citations were approved by the authors and added to the revised manuscript.
[22] Rafiee M, Yang J, Kitipornchai S. Thermal bifurcation buckling of piezoelectric carbon nanotube reinforced composite beams. Comput Math Appl 2013; 66: 1147–60.
[23] Araneo R, Bini F, Rinaldi A, Notargiacomo A, Pea M, Celozzi S. Thermal-electric model for piezoelectric ZnO nanowires. Nanotechnology 2015;26:265402.
[38] Mirzaei M, Kiani Y, Thermal buckling of temperature dependent FG-CNT reinforced composite conical shells. Aerospace Sci Techn 2015;47:42-53.
[42] Fan J, Huang J, Ding J. Zhang. Free vibration of functionally graded carbon nanotube-reinforced conical panels integrated with piezoelectric layers subjected to elastically restrained boundary conditions. Adv Mech Eng 2017;9(7):1-17.
[43] Belal Ahmed Mohamed MS, Zishun L, Kim ML. Active control of functionally graded carbon nanotube-reinforced composite plates with piezoelectric layers subjected to impact loading. J Vib Control 2019, 0:0; 1-18.

Reviewer 2 Report
Review of
“BUCKLING BEHAVIOR OF FG-CNT REINFORCED COMPOSITE CONICAL SHELLS SUBJECTED TO A COMBINED LOADING”
by
Abdullah Sofiyev, Francesco Tornabene, Rossana Dimitri, Nuri Kuruoglu
This work aims at investigating the buckling behavior of FG CNT-reinforced composite conical shells under combined loading conditions. The problem is tackled theoretically according to the first order shear deformation shell theory, whose basic equations are solved in closed form based on a Galerkin procedure. Many numerical examples are here performed by the authors to check for the accuracy of the proposed model, and to study the sensitivity of the mechanical response to different geometrical parameters of the structure, proportional loading parameter, type of distribution and volume fractions of CNTs. The work is well written and organized, and it is worthy of publication, in view of the interesting results and concluding remarks, both from a scientific and practical standpoint. It is just suggested to check for some isolated spelling and/or grammar mistakes, before accepting the work definitively.
Author Response
19.02.2020
EXPLANATION TO REVIEWER 2:
First of all, we would like to thank the highly respected Reviewer 2 for his/her improving remarks and the time spent for them.
Comments and Suggestions for Authors
-Reviewer 2
This work aims at investigating the buckling behavior of FG CNT-reinforced composite conical shells under combined loading conditions. The problem is tackled theoretically according to the first order shear deformation shell theory, whose basic equations are solved in closed form based on a Galerkin procedure. Many numerical examples are here performed by the authors to check for the accuracy of the proposed model, and to study the sensitivity of the mechanical response to different geometrical parameters of the structure, proportional loading parameter, type of distribution and volume fractions of CNTs. The work is well written and organized, and it is worthy of publication, in view of the interesting results and concluding remarks, both from a scientific and practical standpoint. It is just suggested to check for some isolated spelling and/or grammar mistakes, before accepting the work definitively.
EXPLANATOUN: Thank you very much for your constructive and positive comments.
Reviewer 3 Report
In their manuscript, authors present advanced modeling of nanoscale conical shells under combined loading. The topic and the level of modeling is suitable for the special issue "Advanced Mechanical Modeling of Nanomaterials and Nanostructures". I lack the theoretical expertise to technically criticize or validate the details of the theoretical treatment, but the results appear reasonable and are in agreement with previous related simulations. I recommend publishing, provided that the authors address the general criticism:
The introduction could be more aligned with the overall topic of the article, i.e. theoretical modeling. The very beginning of the intro also needs references. Moreover, authors need to rephrase faulty sentences like "...generation of CNTs is strictly related to the creation and evaporation of fullerene..." or "...become hollow pipes when the graphite layer, i.e., graphene, turns into a cylindrical shape," which are not correct. (Graphene indeed can be turned into CNTS [e.g. PRB 85, 085428 (2012), Nature Comm. 4, 2548 (2013)], but that is not the usual way.)
What is "open literature"?
The text needs to be debugged and revised for fluency.
The visual display of images needs to be clarified. Now the images are too messy.
Author Response
19.02.2020
EXPLANATION TO REVIEWER 3:
First of all, we would like to thank the highly respected Reviewer 3 for his/her improving remarks and the time spent for them.
Comments and Suggestions for Authors
-Reviewer 3
SUGGESTION 1: In their manuscript, authors present advanced modeling of nanoscale conical shells under combined loading. The topic and the level of modeling is suitable for the special issue "Advanced Mechanical Modeling of Nanomaterials and Nanostructures". I lack the theoretical expertise to technically criticize or validate the details of the theoretical treatment, but the results appear reasonable and are in agreement with previous related simulations. I recommend publishing, provided that the authors address the general criticism:
The introduction could be more aligned with the overall topic of the article, i.e. theoretical modeling. The very beginning of the intro also needs references. Moreover, authors need to rephrase faulty sentences like "...generation of CNTs is strictly related to the creation and evaporation of fullerene..." or "...become hollow pipes when the graphite layer, i.e., graphene, turns into a cylindrical shape," which are not correct. (Graphene indeed can be turned into CNTS [e.g. PRB 85, 085428 (2012), Nature Comm. 4, 2548 (2013)], but that is not the usual way.)
EXPLANATION 1: Thanks. You are right. But we did not write that the graphene turns into a cylinder in the usual way. Accordingly, we cited Iijima [15]. Adding additional resources will make this better understood. For this reason, we have included and cited the following publications in the revised manuscript suggested by referee (see, Refs [16, 17] in the revised paper).
[16] Kit OO, Tallinen T, Mahadevan L, Timonen J, Koskinen P. Twisting graphene nanoribbons into carbon nanotubes. Phys Rev B 2012;85:085428.
[17] Lim HE, Miyata Y, Kitaura R, Nishimura Y, Nishimoto Y, Irle S, Warner JH, Kataura H, Shinohara H. Growth of carbon nanotubes via twisted graphene nanoribbons. Nature Commun 2013;4: 2548.
SUGGESTION 2: What is "open literature"?
EXPLANATION 2: Some studies and projects related to the defense industry do not appear in the literature. In this sense, we wrote to “open literature”. The word "open" can be deleted here. We deleted the word "open".
SUGGESTION 3: The text needs to be debugged and revised for fluency.
EXPLANATION 3: We have made a careful check throughout the whole manuscript, and tried to identify and eliminate all the grammatical mistakes.
SUGGESTION 4: The visual display of images needs to be clarified. Now the images are too messy.
EXPLANATION 4: The quality of the graphics has been improved.
Reviewer 4 Report
This is an interesting article where the authors study buckling behavior of reinforced composite under combined loading. The proposed solution of the problem is elegant, and the correspondence of the simulation results to experimental data confirms the accuracy of predictions and the significance of the work as a whole. In fact, improving the properties of materials through reinforcement is one of the most relevant topics in modern materials science (metamaterials, heterogeneous materials, architectured materials etc.). The reviewer believes that the article should be published after corresponding following concerns:
1. It would be advisable to avoid using the abbreviation «FG-CNT» in the title.
2. In the introduction, it is necessary to highlight the more general direction of using reinforced composite [O. Bouaziz, H. S. Kim, Y. Estrin, Architecturing ofMetal-Based Compositeswith ConcurrentNanostructuring: A NewParadigm of Materials Design, Adv. Eng. Mater.2013, 15,336–340], as well as methods for their design and production [Estrin, Y., Beygelzimer, Y. and Kulagin, R. (2019), Design of Architectured Materials Based on Mechanically Driven Structural and Compositional Patterning. Adv. Eng. Mater., 21: 1900487.].
Author Response
19.02.2020
EXPLANATION TO REVIEWER 4:
First of all, we would like to thank the highly respected Reviewer 4 for his/her improving remarks and the time spent for them.
Comments and Suggestions for Authors
-Reviewer 4
This is an interesting article where the authors study buckling behavior of reinforced composite under combined loading. The proposed solution of the problem is elegant, and the correspondence of the simulation results to experimental data confirms the accuracy of predictions and the significance of the work as a whole. In fact, improving the properties of materials through reinforcement is one of the most relevant topics in modern materials science (metamaterials, heterogeneous materials, architectured materials etc.). The reviewer believes that the article should be published after corresponding following concerns:
SUGGESTION 1: It would be advisable to avoid using the abbreviation «FG-CNT» in the title.
EXPLANATION 1: Since FG-CNT has become a pattern, it has been used frequently in the literature in recent years. For this reason, it may be more appropriate not to change the title of the manuscript.
SUGGESTION 2: In the introduction, it is necessary to highlight the more general direction of using reinforced composite [O. Bouaziz, H. S. Kim, Y. Estrin, Architecturing ofMetal-Based Compositeswith ConcurrentNanostructuring: A NewParadigm of Materials Design, Adv. Eng. Mater.2013, 15,336–340], as well as methods for their design and production [Estrin, Y., Beygelzimer, Y. and Kulagin, R. (2019), Design of Architectured Materials Based on Mechanically Driven Structural and Compositional Patterning. Adv. Eng. Mater., 21: 1900487.].
EXPLANATION 2: The following studies were examined and added to the introduction and reference section, considering that they would contribute to the manuscript.
- [1] Bouaziz O, Kim HS, Estrin Architecturing of metal-based composites with concurrent nanostructuring: a new paradigm of materials design. Adv Eng Mater 2013; 15:336–40
- [2] Beygelzimer Y, Kulagin R. Design of Architectured Materials Based on Mechanically Driven Structural and Compositional Patterning. Adv Eng Mater 2019;21: 1900487.
- [3] Estrin Y, Beygelzimer Y, Kulagin R. Design of architectured materials based on mechanically driven structural and compositional patterning. Adv Eng Mater 2019;21:

Round 2
Reviewer 4 Report
Accept in present form